# High-Fiber Diet and Crohn’s Disease: Systematic Review and Meta-Analysis

**DOI:** 10.3390/nu15143114

**Published:** 2023-07-12

**Authors:** Victor Serrano Fernandez, Marta Seldas Palomino, José Alberto Laredo-Aguilera, Diana Patricia Pozuelo-Carrascosa, Juan Manuel Carmona-Torres

**Affiliations:** 1Facultad de Fisioterapia y Enfermería, Universidad de Castilla-La Mancha, 45071 Toledo, Spain; victor.serrano3@alu.uclm.es (V.S.F.); juanmanuel.carmona@uclm.es (J.M.C.-T.); 2Hospital Universitario de Toledo, 45007 Toledo, Spain; marta.seldas@hotmail.com; 3Grupo de Investigación Multidisciplinar en Cuidados, Universidad de Castilla-La Mancha, 45071 Toledo, Spain; dianap.pozuelo@uclm.es; 4Facultad de Enfermería de Cuenca, Universidad de Castilla-La Mancha, 16071 Cuenca, Spain

**Keywords:** Crohn’s disease, inflammatory bowel disease, dysbiosis, fiber, diet

## Abstract

Crohn’s disease (CD) is a subtype of inflammatory bowel disease (IBD). CD is a health problem in Western countries such as the US and European nations and is an idiopathic disease; however, certain cases of CD have been associated with intestinal dysbiosis. A systematic review with a meta-analysis was carried out to determine the efficacy of a diet rich in fiber with or without cointervention to improve remission rates for CD. The literature in the PubMed, Scopus, Web of Science, and ClinicalTrials databases was reviewed. The quality of the studies was evaluated using the Johanna Briggs Institute (JBI) scale. This review was conducted in accordance with the structure outlined in the PRISMA statement. In addition, a meta-analysis was performed with a 95% confidence interval (CI) and a random effects model. Eleven studies were included, totaling 2389 patients with CD. Applying a diet rich in fiber with or without the administration of routine therapies improved CD remission rates. Data regarding CD activity, remission time, and adverse effects derived from fiber consumption were analyzed. Consumption of fiber in the diet could improve remission rates for CD patients who receive or do not receive other treatment to maintain remission.

## 1. Introduction

Inflammatory bowel diseases (IBDs) are idiopathic pathologies, among which there are three subtypes: Crohn’s disease (CD), ulcerative colitis (UC), and unclassified IBD (IBD-U), the latter being the least frequent, accounting for 4% of IBD cases in Western countries [1]. Specifically, UC is the most common, accounting for 50% of IBD cases, with CD accounting for 46% among industrialized countries [2,3]. CD presents a diagnostic peak at 15–35 years and is more common among individuals who are under 20 years of age. Regarding the epidemiology, these pathologies show global trends [4]. However, they are more frequent in countries with Western lifestyles, being more common in North America and European countries [5]. In Spain, a study carried out in 2018 showed a prevalence of 0.39% for both IBD types [6]. When comparing the epidemiology between Western and Eastern regions, in North America, the incidence of CD ranges from 8.7 to 10.7 cases per 100,000 inhabitants, and, in Europe, it ranges from 3.6 to 6.3 per 100,000 inhabitants. The incidence and prevalence in Eastern countries have also shown significant increases in recent decades. For example, in Korea, the incidence of CD has increased from 0 to 1.68 per 100,000 inhabitants. Similar results have been observed in other Asian countries, such as China and Taiwan, with significant increases in cases in recent years [7].

In patients with IBD, quality of life is highly compromised compared with that in individuals who do not suffer from gastrointestinal diseases [8]. The signs and symptoms of these diseases are diarrhea, abdominal pain, fever, and complications associated with the malabsorption of nutrients, such as anemia, loss of bone density, and compromised growth in pediatric patients [9].

Despite being a set of idiopathic diseases, several factors that promote the appearance of IBDs have been detected, including the following examples: (1) genetic factors (alterations in the NOD2, ATG16L1, and IL23R genes, etc.); (2) environmental factors related to developed countries (diet, smoking, stress, etc.); and (3) immune system issues, which include decreased intestinal mucus, the infiltration of T and B lymphocytes, and the overproduction of inflammatory mediators such as TNFα, IFNγ, IL-1β, and IL-23, which in turn lead to abnormalities in the action of white cells such as macrophages, neutrophils, and NK T lymphocytes [10].

Another factor to consider in order to understand the pathogenesis of IBDs is the intestinal microbiota. Specifically, compared with individuals without IBD, those with IBD have imbalances in different commensal species at the intestinal level; this phenomenon is known as dysbiosis [11]. In general, a decrease in species with anti-inflammatory properties is observed, and there is an increase in bacteria with inflammatory properties. For example, species such as *Faecalibacterium prausnitzii* (*F. prausnitzii*), *Clostridium leptum* (*C. leptum*), and *Bacteroides* have anti-inflammatory capacities at the intestinal level through the production of short-chain fatty acids (SCFAs), and these species are diminished in patients with IBDs [12]. In contrast, species such as *E. coli* and *Fusobacterium* are more abundant [13].

SCFAs are anti-inflammatory substances that are produced by the degradation of soluble fiber at the intestinal level by the aforementioned microbiota species [14]. Some examples of these substances are acetate, propionate, and butyrate, which have the ability to regulate intestinal inflammation, the immune response, and the composition of the intestinal microbiota [15].

Among IBDs, CD is the most common. To control the symptoms of CD, there are multiple therapeutic options, of which those whose objective is to regulate intestinal inflammation by directly affecting the functioning of the immune system stand out [16]. There are few therapies that focus on regulating the intestinal microbiota, and they are limited to exclusive enteral nutrition [17] and fecal microbiota transplantation [18].

To date, how the consumption of foods rich in fiber can exert beneficial effects on individuals with CD has been studied on other occasions; for example, a Cochrane review carried out in 2019 included 18 studies on dietary interventions for CD. However, not all the studies included in the review analyzed fiber consumption [19]. In 2021, Wagenaar C et al. published a systematic review on the approach to nutrition in various chronic inflammatory diseases. However, it included only two studies that implemented a high-fiber diet for patients with CD [20]. Therefore, it is necessary to analyze the effect of a diet rich in fiber to prevent CD recurrence because the use of such diets has not been thoroughly explored, and there is literature that supports the beneficial effects of fiber regarding the prevention of CD recurrence [21,22].

The objective of this study was to synthesize the scientific evidence available to date of a diet rich in fiber as a method to prevent acute flare-ups of CD as well as to analyze the effects regarding positive clinical improvements.

## 2. Materials and Methods

### 2.1. Design and Information Sources

This systematic review was carried out in accordance with the standards outlined in the Preferred Reporting Items for Systematic Reviews and Meta-Analyses (PRISMA) statement [23]. The details of the study selection process are provided in Figure 1. This review was registered in PROSPERO under registration number CRD42022372891.

The following databases were searched: PubMed, Scopus, Web of Science, and ClinicalTrials.

### 2.2. Search Strategy

The searches were conducted to answer the research question directly, which was developed in the population, intervention, comparison, and outcome (PICO) format (Table 1). The searches were carried out between August 2022 and February 2023.

Components of the PICO research question are as follows: P: problem; I: intervention to be analyzed; C: comparison or control; and O: outcomes.

The clinical question was as follows: in the population with CD, does an adequate consumption of fiber in the diet provide greater benefits than conventional therapies and normal diets to prevent CD recurrence and improve CD symptoms?

The detailed search strategies for each database are listed in Table 2.

### 2.3. Inclusion and Exclusion Criteria

The inclusion criteria used to select the studies that are part of this review were as follows:Observational studies;Clinical trials;Studies that analyzed the consumption of a diet rich in fiber to maintain CD remission;Studies in Spanish or English.

The following types of studies were excluded:Studies that were conducted with animals;Studies that did not include patients with CD in their study population.

### 2.4. Selection of Studies

Study selection was carried out by two researchers, V.S.F. and M.S.P., using the inclusion and exclusion criteria mentioned above. The searches yielded 677 results (the Mendeley computer program was used to avoid duplicates); the selection process followed the flow chart detailed in the PRISMA statement [16]. Once duplicates were excluded, titles and abstracts were read to determine which studies may have met the objectives of the review. Then, those studies were read in full to determine whether they would be included in the systematic review; ultimately, 11 studies were selected. In cases of doubt or discrepancy, a third author (J-M.C.T.) was consulted.

### 2.5. Evaluation of the Quality of the Studies

To assess the quality of the studies and detect biases, Johanna Briggs Institute (JBI) scales were used [24] for each type of study selected. The tool used depended on the type of study. The scores for each study are included in the Appendix A. Specifically, the quality of randomized clinical trials (RCTs) (Appendix A), quasi-experimental studies (Appendix A), cohort studies (Appendix A), and cross-sectional studies (Appendix A) was assessed. There are 4 response options for all the items in these tools: Yes, No, Unclear, or Not applicable. If a result was not applicable in any item, the total score decreased by as many points, as there were nonapplicable items in the study to be evaluated. No eligible study was excluded from the review due to quality.

The quality assessment was independently reviewed by two authors: V.S.F. and M.S.P. The interrater reliability was high, and any discrepancies were discussed between J-M.C.T. and J-A.L.A. until an agreement was reached.

### 2.6. Data Extraction

Data extraction was carried out by two researchers: V.S.F. and D-P.P.C. From each selected study, the following data were collected: (1) title and authors; (2) year and country; (3) study design; (4) sample characteristics: sample size, selection, and age of participants; (5) study objective; and (6) main results: type of fiber consumption and clinical remission rate.

When interpreting the information found, the diet was considered effective when it contributed to maintaining the clinical remission of CD alone or in conjunction with other therapeutic strategies, and the intervention was considered safe as long as it did not produce adverse effects that were different from those produced by conventional therapies.

### 2.7. Analysis of the Data Obtained

For the studies selected for this review, a narrative synthesis was performed. The data were analyzed to compare results between patients who consumed a diet rich in fiber and patients who received the usual treatments or consumed common diets. Additionally, complications derived from consuming diets rich in fiber were recorded.

For the quantitative analysis, a meta-analysis was performed using the inverse variance method of random effects. The standard deviations (SDs) of the pre- and postparameters (high fiber diet + infliximab versus control) of each study were calculated. Statistical heterogeneity was evaluated with the I^2^ statistic. I^2^ ≤ 25%, I^2^ between 26 and 50%, and I^2^ > 50% were used to define low, moderate, and statistically significant homogeneity, respectively [25]. Finally, the effect sizes of all included studies were combined to estimate the overall effect size and 95% confidence interval (CI) and a random effects model. Funnel plots and the Egger’s test were used to verify publication bias. Statistical significance was set at 0.05. The analyses were performed with RevMan software, version 5.4.

## 3. Results

The searches of the four databases yielded a total of 677 results. The Mendeley computer tool was used to eliminate duplicates and manage the references. After eliminating duplicates, a total of 659 results remained.

Following the criteria established, 515 results were excluded. Eleven studies met the aforementioned inclusion criteria [21,22,26,27,28,29,30,31,32,33,34] and were used to conduct the systematic review. Figure 1 shows the flow chart for the selection of studies following the recommendations of the PRISMA statement.

All the selected studies were written in English and were either RCTs or observational studies that investigated diets rich in fiber and from which conclusions could be drawn regarding the maintenance of clinical remission in patients with CD. The characteristics of the included studies are presented in Table 3.

Of the 11 studies selected, 2 were RCTs [32,34], 3 were quasi-experimental studies without a control group [29,30,31], 3 were cohort studies [21,26,28], and the remaining 3 were cross-sectional studies [22,27,33]. Six studies [21,26,27,28,33,34] investigated IBDs, that is, they included UC and IC in addition to CD; the remaining five included only patients with CD [22,29,31,32].

The selected studies included a total population of 5010 subjects, of whom 44% were men and 56% were women. This population ranged in age from 13 to 77 years.

In the studies, there were 2389 patients with CD, and the remaining participants were patients with UC or healthy subjects. The sample included patients with all types of CD based on the Montreal classification (ileal, colic, ileocolic, and upper gastrointestinal involvement) [35].

### 3.1. High-Fiber Diet vs. Normal Diet vs. Gluten-Free Diet

Two of the eleven studies selected compared these types of diets. In general, the high-fiber diet was safer than the gluten-free diet [26]. Additionally, compared with a normal diet, a diet rich in fiber showed greater efficacy, although differences in terms of safety were not assessed [32].

Specifically, the study carried out by Schreiner P, et al. [26] compared the differences between a high-fiber and a gluten-free diet in patients with IBD. In the group of patients with CD, complications related to the disease were less frequent among the subjects who consumed the diet rich in fiber. For this group, the complication rates were 42.4%, compared with 60.5% in the group that consumed the gluten-free diet. In addition, in that same study, the consumption of a diet rich in fiber was associated with greater microbial diversity in all patients with IBD [26]. However, in terms of disease activity, and the need for hospitalization and surgery, no differences were found between diets.

The study carried out by Heaton K, et al. [32] compared the administration of a diet rich in fiber and a normal diet. The results showed a higher hospitalization rate in the group that received the normal diet than in the experimental group (34 vs. 11, respectively), with a statistically significant p value; in addition, the researchers also observed a decrease in the mean number of days of hospitalization for the experimental group (6 vs. 15 days).

**Table 3 nutrients-15-03114-t003:** Results table.

Author/Year/Country	Design	Population	Intervention	Results	Conclusions	Quality of the Studies
Schreiner P, et al. [26] 2019 Switzerland	Prospective cohort study	1313 subjects (pediatric and adult IBD patients belonging to the IBDCS study carried out in 2006 by Pittet V, et al. [36]).	FFQ completed by patients with IBD (from a Swiss cohort with a 9-year follow-up) who consumed a vegetarian and a gluten-free diet.	Uncomplicated CD rate of 57.6% for the vegetarian diet vs. 39.5% for the normal diet; greater complications in subjects who consumed a normal diet vs. those who consumed a vegetarian diet (60.5 vs. 42.4%), with *p* value = 0.039	Changes were evidenced in the microbiota of subjects who consumed a vegetarian diet, but there was no clinical improvement in IBD.	7/11
Tasson, L, et al. [27] 2017 Italy	Cross-sectional study	103 subjects over 18 years of age diagnosed with IBD at least one year before starting the study (50 with active disease and 53 in remission).	FFQ completed by patients with IBD during 1 year of follow-up.	Odds ratio (OR) of new relapses of 1 and similar for Q1–Q3 who habitually consumed fruit and vegetables; an OR of 0.43 was observed for Q4, with a value of *p* = 0.14.	Foods rich in fiber had a protective effect on acute flare-ups in patients with IBD.	8/8
Dolovich C, et al. [28] 2022 Canada	Cohort study	153 participants aged 18–70 years diagnosed with IBD	FFQ questionnaire with 2 years of follow-up.	OR of 0.47 of CD flare-ups in subjects who obtained maximum scores (29–40); OR of 3.63 in the same subjects who obtained scores of 21–24.	There was a positive association between the quality of the diet and the absence of CD flare-ups.	8/11
Chiba M, et al. [29] 2017 Japan	Quasi-experimental trial	60 patients started the protocol, of whom 44 with CD were between 13 and 77 years of age.	Infliximab + vegetable diet for patients with CD for 6 weeks.	Of the 44 patients who consumed a high-fiber diet combined with infliximab, all maintained clinical remission, 84% experienced a decrease in C-reactive protein (CRP), and 46% experienced mucosal healing, with *p* values > 0.05.	Infliximab in combination with a high-fiber diet induced remission in the majority of CD patients.	8/9
Chiba M, et al. [30] 2010 Japan	Quasi-experimental trial	22 CD patients aged 19 to 77 years who achieved remission with infliximab, metronidazole, or surgery at Nakadori General Hospital.	Semivegetarian diet (32.4 g/day of fiber) for patients with CD for 2 years.	Of the 22 patients, 17 completed the follow-up without relapses at two years from the start of the study; 5 experienced relapses.	A semivegetarian diet was safe and effective in maintaining remission in patients with CD, reducing the CRP levels.	6/9
Chiba M, et al. [31] 2022 Japan	Quasi-experimental trial	24 CD patients aged 19–65 years receiving initial treatment with infliximab.	Infliximab + vegetarian diet (32.4 g/2000 kcal) with a 10-year follow-up.	At 4 years, 52% of the subjects were flare-up free; 19% required surgery at 10 years.	Infliximab in conjunction with a high-fiber diet improved long-term remission rates for individuals with CD.	8/9
Heaton K, et al. [32] 1979 England	RCT	64 patients in total: 32 patients with CD (experimental group) and 32 patients in the control group.	Diet rich in fiber (33.4 g/day) for the experimental group and a normal diet for the control group for 5 years.	Average of 11 hospitalizations in the intervention group vs. 34 for the control group, with a *p* value < 0.01; median of 6 days of hospitalization in the experimental group vs. 15 days in the control group, with *p* < 0.02.	A diet rich in fiber improved the prognosis of patients with CD, reducing the need for hospitalizations.	8/13
Mirmiran P, et al. [33] 2019 Tehran	Cross-sectional study	143 patients with IBD, and 32 with CD.	FFQ completed by patients with IBD, with a follow-up of 14 months.	Usual average fruit consumption of 297 g/day by individuals with inactive CD vs. 288 g/day by individuals with active CD, with *p* = 0.51; mean average consumption of vegetables of 190 g/day by individuals with inactive CD vs. 193 g/day by individuals with active CD, with *p* = 0.72.	No significant differences were found between the consumption of fruits and vegetables in relation to CD activity.	7/8
Brotherton C, et al. [21] 2016 USA	Cohort study	1619 adults, of whom 577 with CD were in clinical remission.	26-item survey with a 6-month follow-up.	OR of 0.72 for subjects with CD in Q2 who consumed 13.4 g/day of fiber; OR of 0.57 for Q4 subjects who consumed 23.7 g/day	The consumption of fiber in the diet was associated with fewer recurrences of CD.	7/11
Opstelten J, et al. [22] 2019b Netherlands	Cross-sectional study (diet of IBD patients vs. healthy patients) and longitudinal study evaluating the risk of flare-ups	165 subjects aged 18–70 years with a diagnosis of IBD (participants in the longitudinal part of the study).	FFQ completed by subjects with IBD with a follow-up of 29 months.	OR = 3.65 for a fiber consumption of 21.5 g/day, with a statistically significant *p* value < 0.05.	There was a positive association between high fiber intake and the risk of flare-ups in patients with CD	6/8
Lacerda J, et al. [34] 2021 Portugal	RCT	53 subjects: 25 patients with IBD (experimental group) and 28 healthy controls; of the experimental group, 13 had CD.	Mediterranean diet for 8 weeks and adjusted to caloric needs; in the intervention group, a greater supply of fiber (beta-glucans) was added, and the control group received a normal supply of fiber.	100% of CD patients who consumed a personalized diet were free of flare-ups. In 8 weeks, fiber consumption increased from 21.2 to 30.1 g/day (*p* = 0.01), CRP levels decreased from 5.6 to 1.8 mg/dL (*p* = 0.142), and fecal calprotectin decreased from 470 to 316 μg/g (*p* = 0.47).	The inclusion of specific and personalized nutritional components in the diet was not conclusive regarding producing benefits during CD.	9/13

Abbreviations: FFQ—food frequency questionnaire; OR—odds ratio. Characteristics of the selected studies, effectiveness of fiber consumption in the diet to improve remission rates for patients with CD.

### 3.2. High-Fiber Diet

Of all the selected studies, six assessed the consumption of certain foods together with the risk of relapse or CD activity, of which five showed positive results from consuming a diet rich in fiber [21,27,28,33,34]. However, for one study, the results were different, reporting opposite effects, i.e., linking fiber consumption with higher rates of CD recurrence [22].

The study published by Tasson, L, et al. [27] assessed the habitual consumption of certain foods in patients with IBD and associated these foods with the risk of disease relapse or improvement. They did not observe differences between the appearance of CD flare-ups and the consumption of fruits and vegetables. However, in quartile 4 of the study, the risk of suffering CD flare-ups was estimated at less than half with the consumption of these foods (of 18 subjects, 9 had active disease requiring hospitalization). In that study, the basic treatments used were aminosalicylates (5-ASA), azathioprine, infliximab, and adalimumab.

The study carried out by Dolovich C, et al. [28] evaluated the diets of patients with IBD and qualitatively associated said diets with disease activity. A questionnaire was applied that assessed the habitual consumption of healthy foods, many of which were rich in fiber. The researchers observed an OR for flare-ups of 0.41 for individuals who scored higher on that assessment. However, this study did not specify the basic treatment.

Another study by Mirmiran P, et al. [33] assessed how diet influenced the course of IBD. The mean OR was 0.78 for the risk of IBD flare-ups, as determined using the FFQ score. However, they found no significant association between fiber consumption and changes in disease course.

The study carried out by Opstelten J et al. [22] investigated the relationship between the nutrients that subjects ingested and the rate of relapse. There was a positive association between fiber consumption and flare-ups, with a statistically significant OR of 3.65, showing how fiber consumption increased the risk of recurrence.

However, the study carried out by Brotherton C, et al. [21] yielded results that were opposite to previous findings. The objective of the study was to relate the consumption of fiber in the diet with CD recurrence over a period of 6 months. The OR was 0.72 for individuals with CD who consumed the least amount of fiber daily and 0.57 for those who consumed the highest amount of fiber.

Finally, the trial carried out by Lacerda J et al. [34] analyzed the effects of diet on intestinal permeability and the course of IBD. Of the 13 subjects with CD who consumed a high-fiber diet in the study, all remained flare-up-free and experienced a decrease in acute-phase reactants as the daily fiber intake increased over the 8-week duration of the intervention; however, the decrease in CRP and fecal calprotectin was not statistically significant.

Due to the heterogeneity of the outcome variables in these studies, it was not possible to carry out a quantitative analysis (meta-analysis) of the benefits of a high-fiber diet alone.

### 3.3. High-Fiber Diet Combined with Infliximab

Three studies analyzed how the consumption of a diet rich in fiber influenced patients receiving infliximab as background therapy [29,30,31]. The administration of infliximab together with dietary fiber improved remission rates over time, keeping acute phase reactants under control [29,30] and, in some cases, improving the clinical course of the disease [29,31].

A study conducted by Chiba M et al. [29] observed whether infliximab together with a diet rich in fiber improved remission rates in patients with CD. Of the total number of participants, 96% remained in clinical remission, 84% experienced a decrease in CRP, and 46% experienced mucosal healing 6 weeks after starting the combination interventions. However, the data were not statistically significant.

In another study by the same author [30], 17 of the participants who consumed a semivegetarian diet remained flare-up-free for 2 years, with normal CRP concentrations and a good nutritional status, and 5 participants relapsed during that time. The patients included in this study received basic treatment with infliximab and sulfasalazine, and five underwent intestinal resection.

The third study carried out by Chiba M et al. [31] investigated the total time that patients remained in remission when receiving infliximab and a nutritional intervention. Fifty-two percent of the subjects remained flare-up-free for 10 years. Colonoscopies were performed in 8 of 13 patients in remission, revealing a totally healthy mucosa in 4, mild ulcers in 3, and active ulcers in 1 patient with high levels of CRP.

The results of the meta-analysis (Figure 2) showed a beneficial effect (a decrease in the mean Crohn’s Disease Activity Index (CDAI) at 6 weeks) in the group of patients in which a high-fiber diet was administered together with infliximab (MD −1.4; CI 95% −1.7, −1.1; *p* = 0.08; I^2^ = 0%).

### 3.4. Safety of a High-Fiber Diet for Individuals with CD

Of all the selected studies, only three showed any adverse effects. Two patients in the study by Chiba M et al. [29] had intestinal obstruction attributable to dietary fiber. In another study [32], one subject experienced sacroiliitis. However, this condition could not be attributed to fiber consumption.

Notably, another study [22] reported that the administration of dietary fiber at a dose of 21.3 g/day was associated with higher rates of CD relapse.

However, in the remaining studies analyzed [21,26,27,28,30,31,33,34], no adverse effects occurred from the administration of dietary fiber.

## 4. Discussion

Based on the results obtained, high fiber intake through the diet is not only safe but also effective in maintaining remission in conjunction with other therapies for patients with CD.

From a nutritional point of view, certain habits increase the risk of IBD flare-ups; for example, consuming red meat more than four times a week increases the risk of UC flare-ups [26]. In the study carried out by Schreiner et al. [26], lower microbial diversity was observed in CD patients who consumed meat more than four times a week, potentially posing a risk of suffering a flare-up of this pathology [26]. However, from the nutritional perspective, fiber consumption in this group of patients was associated with lower rates of hospitalization, although with no signs of clinical improvement in CD [25]. In addition, a high intake of saturated and monounsaturated fats and higher levels of omega 6 fatty acids than omega 3 fatty acids are associated with a higher risk of CD relapses [32,37,38].

Although the efficacy of fiber in maintaining CD remission cannot be denied [21,28,31,32], the amount of fiber consumed was not specified in all studies. In five studies [21,30,31,32,34], a specific amount of daily fiber consumed by the subjects was specified. In two studies [30,34], favorable but not statistically significant results were reported. Notably, another study included in this review [22] found that fiber consumption was associated with a greater number of CD flare-ups at doses of 21.3 g/day. However, of the 11 studies included, it was the only one that reported that the consumption of fiber in the diet is a detrimental factor for patients with CD. This could be because the study evaluated food consumption for an insufficient time to assess the nutritional intervention correctly, because of errors made by the subjects when completing the questionnaire, or because of a definition of clinical remission that was dependent on many factors. These results could also be due to the association between fiber consumption and obstructive symptoms at the intestinal level, which was not investigated in the study.

By professional consensus, the consumption of fiber is recommended to all patients with IBD, as long as they do not have intestinal stenosis; insoluble fiber could act at the level of the intestinal lumen, producing an increase in fecal waste and increasing the risk of occlusive symptoms at the intestinal level [39,40]. Likewise, fiber in the diet is effective for maintaining remission in patients with CD, and other studies have shown that a diet rich in fiber is effective for maintaining remission in patients with UC, another type of IBD [19,20,41,42,43]. Some articles [19,44] have concluded that a diet rich in fiber maintains remission in patients with CD, and these findings are consistent with the results obtained in this review [21,27,28,29,30,31,32].

A Cochrane review reported how a diet rich in fiber produces an increase in SCFAs, promoting intestinal homeostasis and helping to control inflammation in patients with CD [19], as specified in the results of a study collected in this review [34]. Notably, in this study, levels of CRP and fecal calprotectin remained stable, and there was no statistically significant association between dietary fiber consumption and the course of CD. However, that Cochrane review evaluated how the consumption of dietary fiber helps to induce remission [19], whereas this review analyzed how fiber helps maintain remission. Therefore, there is a need for more studies on the maintenance of clinical remission in patients with CD.

The consumption of dietary fiber by patients with CD was lower than that by healthy controls (4 g/day difference between groups), with a total fiber consumption of 14 g/day in patients with CD [43]. However, several studies have reported that low-dose fiber consumption is not associated with an increase in the severity of the pathogenesis of CD [43,44], and these results are supported by studies included in this review [26,33,34]. Other types of diets studied for the treatment of IBD have been exclusion diets, with results showing no greater efficacy in treating IBD than that of diets rich in fiber [20,44].

Fiber administered at controlled doses and in the form of a supplement is effective in controlling the symptoms of CD [41,45]. Specifically, fructans, as a supplement at doses of 15 g per day, have been shown to reduce the Harvey–Bradshaw index in patients with CD and produce an increase in Bifidobacteria in the intestinal lumen of patients [45,46]. In addition, the consumption of fiber in the form of fructooligosaccharide prebiotics reduces CD activity (reduction in the Harvey–Bradshaw index from 9.8 to 6.9 points) [45,46] and reduces susceptibility to developing CD [42]. Fiber consumption is shown to be effective for improving CD symptoms, which is supported by the results obtained in this review [21,27,28,29,30,31,32]. However, in the studies analyzed in this review, the exact amounts of fiber consumed by the subjects were not observed.

To clarify more accurately whether a nutritional intervention rich in fiber is adequate to maintain remission in patients, more RCTs are needed that compare control groups with conventional therapy groups and experimental groups and that specify the diet administered to the patients as well as the amount of fiber needed. Furthermore, such studies should be conducted for a sufficient time to ensure the reliability of the results obtained. Only two studies evaluated interventions long enough to identify the remission rates for patients clearly; the evaluation time was greater than 2 years [30,31].

Among the studies included in the analysis in this review [29,30,31] are those that show that infliximab is effective in maintaining remission in CD patients. However, only in the studies carried out by Chiba M et al. [29,30,31] was the effect of infliximab combined with dietary fiber on maintaining remission analyzed.

### 4.1. Limitations and Strengths of the Review

Among the limitations of this review are the small number of RCTs included in the review and the lack of nutritional specifications of the diets used or investigated in the studies. In the studies included in this review, the type of fiber consumed by the subjects was not accounted for except in one study, which included soluble fiber in the participant diet [34]. This may be because most of the studies included in the review were observational and based on the administration of questionnaires. Only five studies specified the daily amounts of fiber consumed by the participants [21,30,31,32,34].

Another limitation was the heterogeneity in the variables resulting from the included studies; this heterogeneity made it impossible to carry out the meta-analysis. In fact, in the meta-analysis conducted, diet interventions were different among selected studies (vegetarian, semivegetarian, etc.), so it is difficult to compare them. Also, it is important to mention that these studies were low quality, and only two RCTs had a control group [32,34].

Furthermore, one study did not use analytical, or endoscopic criteria to measure CD activity [21]. It is necessary for evaluations of CD status to be carried out by means of specific analytical values such as fecal calprotectin or C-reactive protein, in addition to the health status of patients identified through general examinations.

As strengths, the sample size in each study was sufficient to clearly reach the conclusions obtained. The studies were carried out in different parts of the world; therefore, information was collected from different populations.

### 4.2. Implications for Practice

It seems that high dietary fiber intake can increase the remission time in patients with CD. It is a simple nutritional practice without risk for patients. However, adjuvant treatment with a diet rich in fiber should be supervised by a health professional because one of the functions of health personnel is to provide health education. Under these conditions, the consumption of a high-fiber diet is recommended.

## 5. Conclusions

Dietary fiber intake could increase remission rates in patients with CD who achieve clinical remission with conventional therapies. Due to scarce evidence about the relationship between fiber intake and CD, we cannot recommend a specific dietary intervention. However, it can be stated that a fiber intake between 13.4 and 33.4 g/day is safe in CD patients. Also, a high-fiber diet should be an adjuvant treatment in conjunction with other therapeutic strategies to maintain remission.

More studies are needed to evaluate the amount and type of fiber to improve remission rates for CD. Specifically, we believe it is necessary to conduct more RCTs that analyze conventional therapies and high-fiber diets in combination with conventional therapies, with analytical and endoscopic follow-up to determine the efficacy of the intervention.

## Figures and Tables

**Figure 1 nutrients-15-03114-f001:**
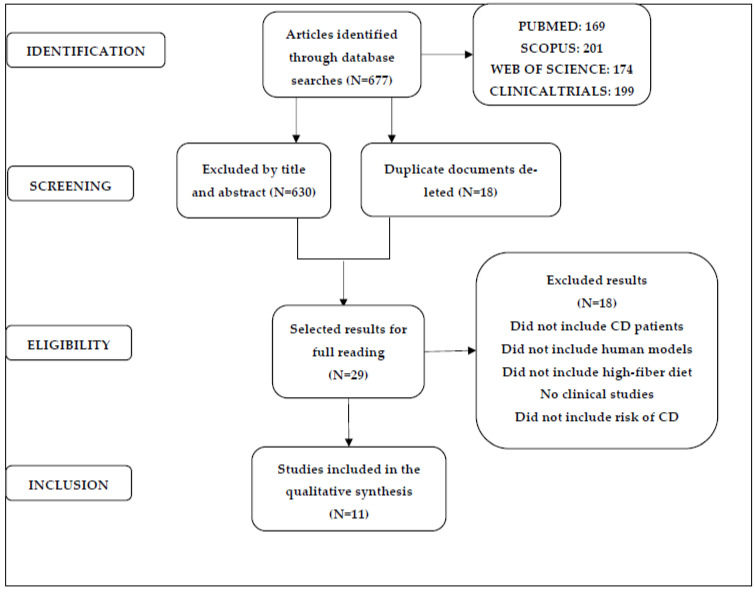
PRISMA flowchart.

**Figure 2 nutrients-15-03114-f002:**
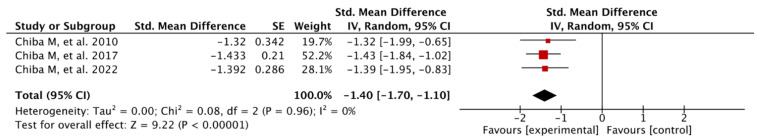
Meta-analysis of the consumption of a high-fiber diet in combination with infliximab [29,30,31].

**Table 1 nutrients-15-03114-t001:** PICO question.

Population	Intervention	Control	Result
Patients with CD	High-fiber diet with or without cointerventions	Conventional therapies (corticosteroids, immunosuppressants, and biological therapies) or other diets	Prevention of flare-ups of CD and improvements in the disease with a diet rich in fiber

**Table 2 nutrients-15-03114-t002:** Detailed searches.

Database	Search String	Filters
PubMed	(Inflammatory bowel disease OR IBD OR Crohn’s disease) AND (dietary fiber OR dietary fiber OR diet) NOT (exclusion diet) NOT (FODMAP) NOT (Western diet) NOT (Mediterranean diet) NOT (enteral nutrition) NOT (elemental diet)	“Clinical study”, “clinical trial” and “observational study”
Scopus	TITLE (“inflammatory bowel disease”) OR TITLE (“IBD”) OR TITLE (“Crohn’s disease”) AND TITLE (“dietary fiber”) OR TITLE (“dietary fiber”) OR TITLE (“diet”) AND NOT TITLE (“exclusion diet”) OR TITLE (“FODMAP”) OR TITLE (“Western diet”) OR TITLE (“Mediterranean diet”) OR TITLE (“enteral nutrition”) OR TITLE (“elemental diet”)	Language (English and Spanish) and type of document (articles)
Web of Science	(((((((((TS=(“inflammatory bowel disease”)) OR TS=(“IBD”)) OR TS=(“Crohn’s disease”)) AND TS=(“dietary fiber”)) NOT TS=(“exclusion diet”)) NOT TS=(“FODMAP”)) NOT TS=(“Western diet”)) NOT TS=(“Mediterranean diet”)) NOT TS=(“enteral nutrition”)) NOT TS=(“elemental diet”)	None
ClinicalTrials	(inflammatory bowel disease OR IBD OR Crohn’s disease) AND (dietary fiber OR dietary fiber OR diet) NOT (exclusion diet) NOT (FODMAP) NOT (Western diet) NOT (Mediterranean diet) NOT (enteral nutrition) NOT (elemental diet)	None

Databases, search strings with Boolean operators, and MeSH terms and filters used.

## Data Availability

The data presented in this systematic review are available in this paper and its Appendix A.

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
