# Peer review of "High-Fiber Diet and Crohn’s Disease: Systematic Review and Meta-Analysis"

_nutrients, 2023, doi:10.3390/nu15143114_

Round 1
Reviewer 1 Report
Dear authors an interesting subject and a interesting work.
Was it possible in some of the selected trials to indentify the type of fiber, soluble vs insoluble ?
Could you please improve the presentation of the table 3 and the figure 2 ?
Reviewer 2 Report
- - “indeterminate colitis (IC)” or unclassified IBD (IBD-U)?
- - “indeterminate colitis (IC), the latter being the least frequent, accounting for 7-10% of IBD cases worldwide. Specifically, CD is the most common, accounting for 25-30% of IBD cases, with UC accounting for 20%”
20 + 30 + 10 = 60% -> what about the remaining 40%!?
- - Use always upper case for western and eastern
- - Use oxford comma in the whole text
- - Define the abbreviations the first time they appear (for example, “F. prausnitzii”)
- - Write the abbreviations of the authors that performed the studies’ evaluation instead of XXX
- - Why did you choose the random effects in the meta-analysis?
- - Graph of Figure 2 in wrong: in fact, the squares intersect the vertical line of the “0” values, despite the confidence intervals don’t intersect the 0 value. In addition, write what is the outcome
- - Were CD patients excluded from the studies if they were affected by stenosing phenotype?
Moderate revision is needed.
Round 2
Reviewer 2 Report
Thank you for the corrections
It is improved
Author Response
Thank you very much for giving us the opportunity to improve the manuscript.